# Changes in the 19th Century Cultural Landscape with Regard to City Rights in Western Poland

Dariusz Lorek [1] and Tymoteusz Horbiński [2,*]

[1] Landscape Ecology Research Unit, Faculty of Geographical and Geological Sciences, Adam Mickiewicz University, 61-712 Poznań, Poland; kubal@amu.edu.pl
[2] Department of Cartography and Geomatics, Faculty of Geographical and Geological Sciences, Adam Mickiewicz University, 61-712 Poznań, Poland
[*] Correspondence: tymoteusz.horbinski@amu.edu.pl

**Abstract:** This research study focuses on determining the spatial transformations taking place in selected areas in the context of administrative changes in the 19th century (in the context of city rights) using the example of three neighboring places in western Poland. The occurrence of both individual topographic features and the transformation of structures and spatial relations occurring in the studied area since the 19th century were considered. The source material included archival cartographic studies from six time periods and contemporary data resources. A significant part of the research concerned the development of the possibility of using and presenting the data in an interactive form. The most important functions include comparing three neighboring places at the same time. Programming activities focused on the implementation of all collected archive data in the form of rasters and the construction of a map service divided into three windows (taking into account the turning on of layers simultaneously for all windows). The Leaflet library was used to create the proposed map solution.

**Keywords:** cultural landscape; map service; 19th century; city rights; archival maps

## 1. Introduction

The technology and methods of landscape mapping have evolved over a very long period of time. The oldest record may be Pliny's work (77 AD), Naturalis Historia [1]. In the next phase, the development of numerical approaches increased the precision of the mapping process [2]. A few centuries later, colonization by Europeans contributed to the representation of the "other world" outside the continent. However, it is important to remember the activities of cartographers on the continent. Significant advances in the accuracy of spatial representation on maps occurred at the beginning of the 19th century. Methods of obtaining data from the field evolved, and better and better measuring instruments were used, replacing the earlier non-cartometric methods (e.g., à la vue). An example of this is a series of Prussian topographic maps produced using plane table surveying [3]. The sheets, which were published for half a century from the second decade of the 19th century, were called Urmesstischblätter. A continuation of this series were the Messtischblätter, which were based on more accurate measurements and which depicted conditions from the 1870s onwards, for which several updated editions were produced [4]. Both collections are very well suited for the study of old landscapes and their changes. They are characterized by the same scale of 1:25,000, the same division into sheets and a similar system of conventional symbols (legend) [5].

In the context of modern research, maps have become a part of a cultural heritage that aims to represent past human civilization through map research. In many recent studies, map archives are the main source for analysis. Andrade and Fernandes [6] interpreted maps as an irreplaceable primary source of geographical and political information from the past. Ekim et al. [7] extracted valuable information on transportation infrastructure

and the spatial distribution of settlements for quantitative and geometric analyses. This spatial information is crucial for facilitating decision-making in environmental management [8]. Historical maps can be used in environmental policy, for example, to calculate environmental changes such as elevation changes and land use changes, as Tortora et al. [9] have noted.

As in the cited articles, we use a cartographic research method [10] in which maps are the basic source of information. Therefore, the degree of accuracy (detail of information) is related to the scale of the cartographic material used. The chosen procedure is also based on methodological principles for the use of old maps in research [11,12]. Accordingly, the information obtained was verified and supplemented using information from other (descriptive) source studies.

For this landscape research, the activities related to human decisions in the context of the natural environment and the interventions in its management form the basis for analyzing the changes taking place. The research is based on recording the state of the space in the context of economic transformations (both administrative and changes related to the processes of urbanization and industrialization in 19th century Europe). This approach to change enables the analysis of the cultural landscape of selected areas [13–16].

The research methodology applied in this study, which is based on the use of graphic source material (maps), made it possible to view the cultural landscape in the context of specific spatial objects marked on the map [13,17]. In this context, we can consider the "cultural topographical landscape", which specifies and refines the traditional definition of landscape to the level of landscape information that can be obtained from a topographical map [13,18].

The aim of the study was to determine the extent to which the administrative status of the city (with or without city rights) can influence the state of preservation of or the change in the landscape. We considered whether it is possible to show the relationship between the degree of urbanization processes and spatial changes and the status of a particular unit over a period of almost 200 years. The analysis included neighboring areas representing different types: a village, a town, and an area that had lost its city rights. To facilitate the analysis and interpretation of the changes, a map service was created.

## 2. Materials and Methods

### 2.1. Study Area

Three towns in western Poland were selected as part of the research area. They are located a few kilometers apart and are connected by county road no. 409 (Czerniejewo–Żydowo–Niechanowo). They differ in terms of their administrative category (town, village) and the changes in this respect during the period under consideration. This was the factor that determined their selection for this study. Taking into account the above criteria, the extent to which changes in spatial planning (development of the cultural landscape) are related to changes in the status of individual towns was analyzed. The following administrative units were selected:

- Niechanowo is an example of a village in the list that did not change its status, i.e., from 1830 to 2023, it remained a village;
- Żydowo is a town that changed its status during the period in question, i.e., it had city rights until 1869 and then became a village. Żydowo was a city in the years 1752–1869, and the loss of city rights was connected with economic changes in the second half of the 19th century. As a result of the new territorial divisions in Europe after the Congress of Vienna in 1815, there were changes in individual economic sectors. Textile manufacturing that had been well developed at the turn of the 18th and 19th centuries was adversely affected in Greater Poland under Prussian rule. The number of factories declined, and the towns in which craftsmen had previously worked in this industry became depopulated. This eventually led to a change in status and the loss of city rights for Żydowo;

- Czerniejewo is an example of an urban area that also changed its status during the period under consideration, but this was dictated by the actions of the occupying forces during the Second World War, which had been ongoing since 1939. The change in the town's status was therefore not dictated by factors related to a change in the population or area. Until 1939, Czerniejewo was a town, and at the beginning of the war, it was included as a rural municipality. After the end of the war in 1945, the ministry was asked to resume city rights, which it did in April 1946.

### 2.2. Source Materials

The research included cartographic sources from six time periods. Five of these were topographic maps at a scale of 1:25,000, while the last study was a data source from OSM (OpenStreetMap). For the study, it was important to collect source material at the same level of detail in order to identify real changes related to the occurrence and change in individual topographic features. Moreover, such a research scheme results from the methodology of studying old maps [11,19].

In addition to the collected image material, monographic studies on individual areas were a valuable source of information, which also served to verify and supplement the information contained in the map. A literature search revealed that there are only a few studies that take a comprehensive approach to the history of individual towns. Some of the publications only concern specific objects or only describe a short period of time [20]. Many of them are written memories of people who used to live in the areas [21]. There is a lack of scientific studies that precisely characterize the ongoing changes in individual settlement units. Nevertheless, the above-mentioned descriptive source materials, combined with the content recorded in cartographic studies, make it possible to reconstruct the landscape of past eras. This research approach makes it possible to deepen the knowledge about individual areas, especially in connection with spatial transformations visible at the topographical level.

The selection of time periods was primarily determined by the publication dates of the individual maps. The first two maps from the 19th century, which depict the situation in the years 1830 and 1888, show the greatest time gap, which is mainly due to the fact that there are relatively few studies from this period. The next largest time span includes maps from 1977 to the most recent data from 2023, which is due to the lack of 1:25,000-scale analog map editions in the period mentioned (there are 1:50,000-scale editions).

Using source materials (graphic and descriptive) relating to specific epochs, the spatial changes that took place in the areas studied were analyzed in the context of the cultural landscape. The occurrence of selected objects in terms of qualitative and quantitative diversity was taken into account. Working with archival maps also made it possible to analyze and compare entire spatial structures and to indicate the direction of changes that took place in specific periods.

The process of obtaining and processing archival data into a form that enables their implementation in the map service was the subject of previous research [22].

An important topic when working with archival maps is georeferencing (georeferencing of raster images). The results achieved in this phase (matching accuracy) have a considerable influence on the implementation of further steps in connection with the processing and digital use of archive data. The process of georeferencing old maps requires an individual approach to each sheet in terms of the selection of control points (their number and distribution) or the use of a function responsible for image transformation [23–27].

The Qgis 3.28 program was used to implement the process of adapting the rasters to the coordinate system. The maps were registered in the WGS-84 coordinate system (EPSG 4326). Image registration was performed using the Polynomial 1 function based on four control points. Due to the incomplete cartometry of the oldest map, a larger number of control points were used for its georeferencing [22,24].

Subsequently, individual map sets were characterized, as well as aspects related to their selection and procurement. In this study, the authors used the methodology developed

at that time, modifying selected activities related to the specific nature of the topic under discussion. The difference was in the way the archived information was made available on an online map. In the research from 2020 [22], objects from individual maps (road networks) were vectorized and made available to the website user via individual layers. The created map service presents information about the old landscape in the form of grids of archived maps embedded in the modern OpenStreetMap resource.

*2.3. Map Service*

An important part of the research presented was the development of the map service (Figure 1)—http://th31483-gamescartography.home.amu.edu.pl/ (accessed on 12 May 2020). The authors wanted to explore the topic in the context of the possibilities of using and presenting data on the cultural landscape since the 19th century. The developed product is both the result of the work carried out and, at the same time, a proposal for a methodology for studying the past landscape and current changes. The Leaflet.js library was used for the development [22,28,29]. The map was based on 18 raster maps for the three towns analyzed. Three separate map windows were created, which were synchronized using a layer selection field so that an analyzed state of the cultural landscape could always be seen in three windows. The raster implementation is based on the basic part of the Leaflet.js library code, i.e., L.imageOverlay for two alignment points (top left and bottom right). Each window is blocked so that only the thematic area of the maps is visible. Two overview maps showing the analyzed cities at a smaller scale are also included. They are created as separate objects and have no connection to the maps. The current state is displayed on a layer of vector tiles for OpenStreetMap. This background always appears below other implemented rasters. In order to be able to compare the analyzed state with the current state, a slider was also programmed that changes the opacity for the activated raster. In this way, it is possible to detect and analyze changes in the cultural landscape between two states and three cities simultaneously. The map service does not provide for the overlapping of layers between historical states. The advantage of such a map service is undoubtedly the easier access to map information [30]. The entire work is based on open-source technologies and data, which will make it easier for others to copy the measures used to create this map service.

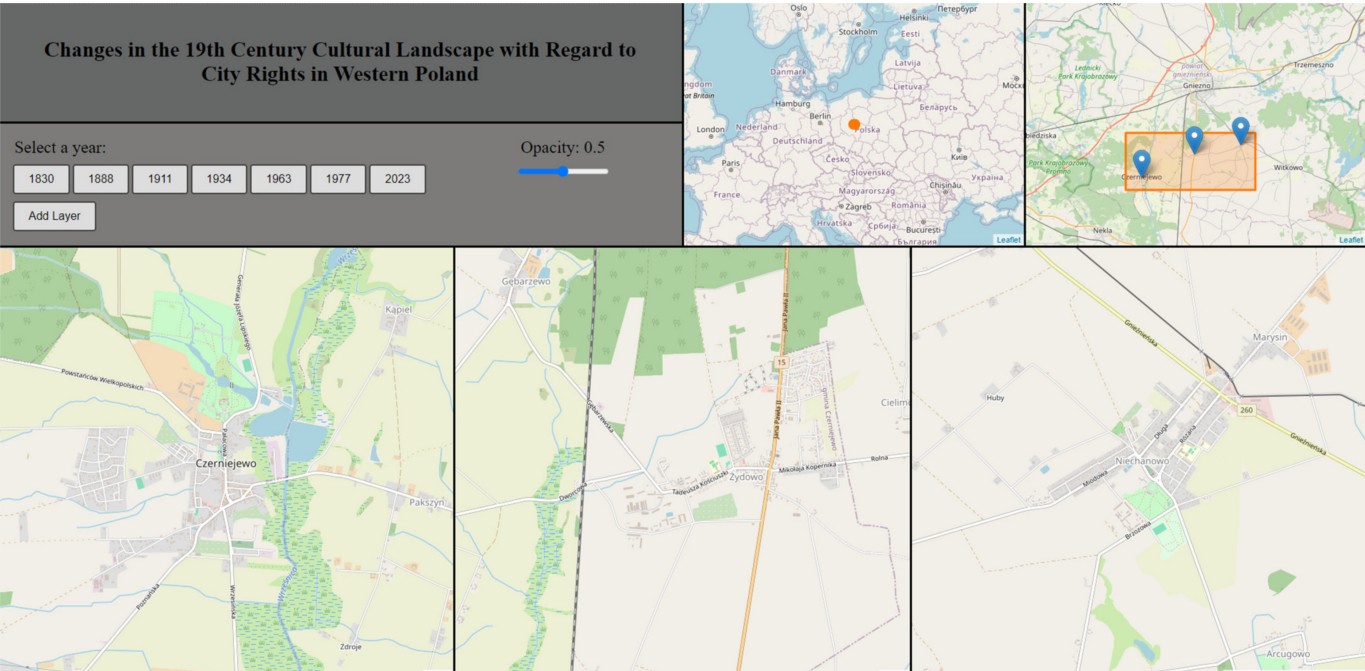

**Figure 1.** Example of the view of a map service.

*2.4. Historical Background*

The changes and interdependencies that took place in the 19th century between economic sectors such as agriculture, crafts, industry, trade and transport are reflected in the individual elements of the topographical landscape of the former Greater Poland, which are recorded in cartographic archive material. By looking at the components of the Greater Poland landscape, it is possible to identify changes and spatial relationships that characterized the individual environmental conditions in the 19th century [13,31]. By comparing the environmental data recorded on maps and other sources, it is possible to classify the characteristics of the space into the economic states under consideration [32]. Cartographic materials perceived in this way represent a source of information in two respects: the occurrence of individual objects—changes in their qualitative and quantitative proportions—and the relationships between objects—the spatial distribution of objects.

The beginning of industrialization in Greater Poland dates back to around the middle of the 19th century, when certain elements appeared in the landscape or existing forms, and spatial structures were changed, usually due to the mechanization of production and transport. From a cartographic point of view, these are individual topographical features that can be depicted on maps and plans, e.g., roads, railroads, surface waters, forests and development areas, but also selective features such as mills, brickworks and raw material extraction sites. In the context of research into changes in the 19th century, the importance of settlement structures and road networks, in particular, for the representation of the earlier state of the environment is pointed out [33]. The changes recorded on the maps show both differences in the quantitative aspect of the phenomenon (increase, decrease) and qualitative changes, e.g., new categories of roads, new types of building materials, which constitute new classes of objects on the map.

On the basis of the cartographic material collected since 1830, it was possible to study the changes in the cultural landscape of Greater Poland. Since there are relatively few monographic studies that comprehensively describe the transformation of individual regions or selected towns, the selection of suitable cartographic material (scale and time) is a valuable way of proceeding. In addition, the use of tools and technologies related to the programming of interactive map services in the whole process enables the implementation of archival sources and their study and presentation in new perspectives.

## 3. Cultural Landscape of the Research Area—Analysis

*3.1. Niechanowo*

The first of the localities under consideration is the village of Niechanowo. The description of the research began with this area because it was the only village out of the three that did not show any changes in the status of its administrative category. Therefore, among the factors that could potentially influence further transformations of the 19th century cultural landscape, issues related to the change in the formal rank of the town can be excluded.

The selection of the plane of "1830" and the setting of partial transparency showed the old spatial structure of the village against the modern background (Figure 2). The landscape marked on the 1830 map shows the typical road–village character of Niechanowo, which developed along the main road connecting Gniezno and Witkowo. The main axis is a communication corridor running through the village, with buildings on the western side. At the end of this road, on the southwestern side, there is a church, and the entire former village complex is completed by the manor house with a palace and a large park. The map also shows two outbuildings of the palace, as well as small water reservoirs (ponds) nearby on the western side. There are farm buildings on the northeastern side of the park. There is also a network of small field paths, some of which are straight and regulated. The relief of the terrain depicted using the method of hachuring shows a series of hills to the northwest. According to the descriptions, the church, rectory and cemetery were located on a small hill [21]. In general, Niechanowo and the neighboring areas were surrounded by plains and

woodless areas in the first half of the 19th century. Outside the village, to the west, there were two windmills on hills, and to the east, there was a farm on the road to Witkowo.

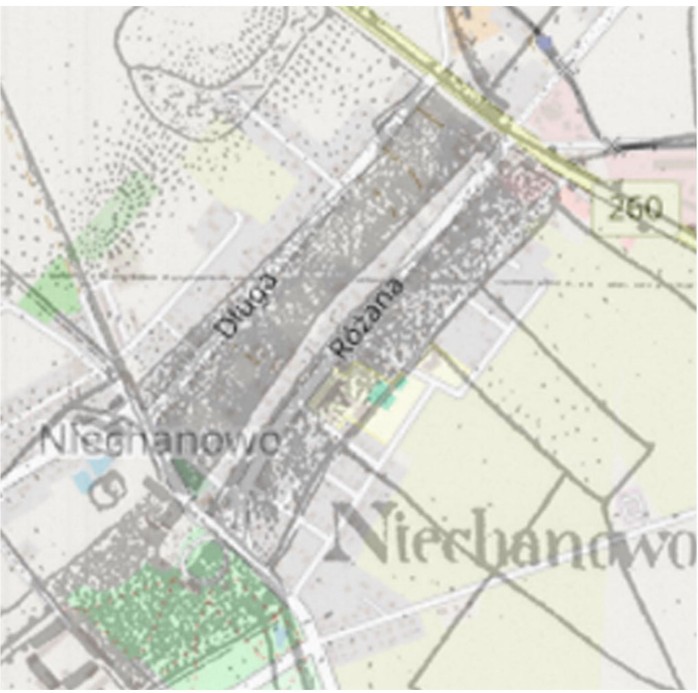

**Figure 2.** Status from 1830 against the background of contemporary data.

Changing the browser window and selecting the layer with the status of "1888" illustrates the differences to the status of 1830, particularly in the way it is presented. This is due to the considerable amount of time that has passed between the publication of the two maps (the method of data collection and the presentation on the maps), namely over half a century. For example, the way in which the terrain is depicted has changed, which has also improved the readability of the map. However, as far as changes in the landscape are concerned, the current structure has not changed significantly. An important feature that was created in the second half of the 19th century and that has been preserved in the town's landscape to this day is the second road that runs parallel to the main traffic axis through the village. There is also a new road section connecting the southeastern borders of the village with the main Gniezno–Witkowo road. As far as road connections are concerned, some roads have been straightened and their status changed from dirt roads to paved roads.

Another important element of the cultural landscape in the second half of the 19th century were the railroad lines. As can be seen in Figure 3, the tracks ran from the northwest and changed direction to the south near Niechanowo. It was a narrow-gage railroad connecting Gniezno, Niechanowo and Mielżyn, and its operation was mainly focused on the transportation of goods and also of passengers. However, the fact that the railroad connection was built does not indicate that the development of the areas in the immediate vicinity of the line on the 1888 map had changed. A new establishment in the area was Marysin Farm, located on the north side of the railroad. According to the literature, it was established in 1847 [21]. It was associated with new crops, mechanization and the general development of agriculture. Not far away was a stream that no longer exists in the modern landscape. Apart from the processes associated with industrialization and urbanization, which took place mainly in the cities, the second half of the 19th century was also marked by progress in agriculture. This manifested itself, among other things, in mechanization, soil improvement and the creation of new farms [21]. The castle farmstead developed noticeably—its area increased and new buildings were added. New buildings

were also built in this part of the village near the southwestern border. The small farm on the road to Witkowo is no longer shown on the analyzed map.

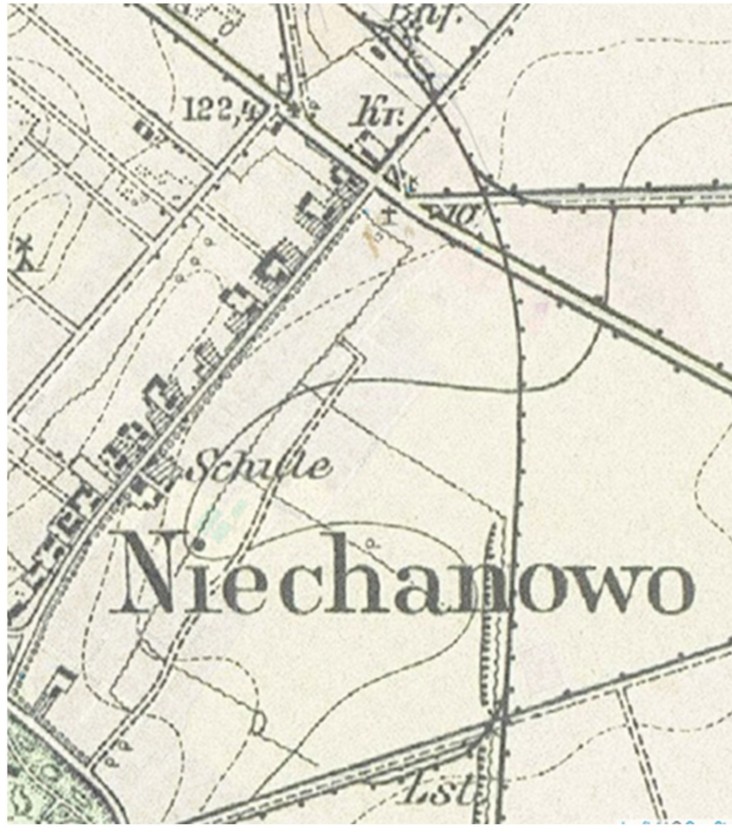

**Figure 3.** Layout of railway routes on the map from 1911.

In addition to other objects that characterize the cultural landscape, the map also shows the location of the windmill in the western part of the village (on a hill). According to Szymanowska and Kozanecki [21], there were two windmills in the village at the beginning of the 20th century. In addition, the place of sand extraction in the western part of the village was marked on the map with the abbreviation "Sgr.".

The next time level is the year "1911". The changes visible here compared with the previous state are minor, and this applies to all the areas under consideration. This is due to the relatively short time interval between the sheets in question and the fact that this is another edition of the same series of Prussian topographical maps. The authors have decided to integrate them into the website because they represent very good comparative material and contain details (objects) that demonstrate the changes in the cultural landscape.

First of all, the expansion of the railroad lines is visible. Two new branches were created in Niechanowo: east to Witków and on the further route toward the distillery. A network of connections between the individual goods was created, which was mainly used for the transportation of agricultural products. The content of the 1911 map shows the second track and the station built in Niechanowo. You can also see that new buildings were constructed near the railroad line. The location of the tavern near the intersection of roads and tracks ("Kr.") was already marked on the earlier map. The further development of Marysin Farm can be seen in the new buildings. A new two-storey school building was built in the central part of the village (apart from the graphic signature, it was also signed on the map). Its construction was a manifestation of the Germanization of Polish society, but both after regaining independence in 1918 and after the wars, it still serves to educate young people [21]. A new feature in the existing landscape of the village and its surroundings is a small area of mixed forest that has been created next to the palace park on the southeastern side.

The landscape from 1934 marked on the map hardly differs from that of the previous year. Some markings of individual objects (signatures) have disappeared, e.g., the signature of the windmill, the school and the inn. However, it should be taken into account that this is a completely new study—a Polish map created by the Military Geographical Institute (English for Wojskowy Instytut Geograficzny = WIG) after regaining independence. Therefore, the way of classifying and marking individual objects may differ, even if the content of the Prussian maps was referred to when creating these sheets [22]. When looking at individual landscape elements recorded in this material, only changes in the way the content is presented are visible. Both in the case of individual objects and entire structures, changes that are significant for landscape research are difficult to recognize, which becomes clear when you switch the windows between the years 1911 and 1934 on the website.

The map from 1963 is also characterized by a certain peculiarity, namely a period in the history of Polish cartography when censorship on maps was in force. It is a map published in the "1942" coordinate system, in which military maps were created after the Second World War. The standards imposed by the USSR (which significantly influenced the Polish authorities at the time) required the publication of separate maps for the civilian population, which were characterized by, among other things, impoverished content and censorship. If one compares the village landscape with the earlier map, there are no significant differences in spatial management apart from the changed graphic representation. Of note is the reduced importance of the road that runs parallel to the main road through the village and the new woodland planting in the southern part of the village next to the farm. Despite the fact that many lands were parceled out after the Second World War, these changes in the structure of the landscape are not visible at the 1:25,000 level of detail.

The last analog map from 1977 shows a more developed Niechanowo both on the eastern side of the main road in the village (where a school was previously built) and to the northeast. New buildings were being constructed at the junction of main roads and railways and along the road to Marysin Farm. The area of the farm had increased and the ownership structure had also changed—it had been converted into a state agricultural farm. The siding leading to the distillery in Niechanowo had disappeared from the area. On the southern and western sides of the village, small streams, water reservoirs and areas previously designated as wetlands had also disappeared.

The present-day landscape of Niechanowo ("2023") is characterized by the absence of part of the railroad line, which ran southwards, while trains do not run on the remaining sections. Completely new buildings were built in the southwestern part. The expansion took place in the area of the former Marysin farm, and several new roads were built in the central part of the village where construction took place. Apart from the changes mentioned above, the 19th century village landscape has largely been preserved in its modern spatial arrangement. Individual objects associated with the economy of past eras, such as windmills, sand quarries or an extensive network of railroad lines, have disappeared, but the former structure of the village, based on the main street (and later two roads) and ending in a large park with a palace and a farm, has remained unchanged.

### 3.2. Żydowo

The year of publication of the oldest map used for the study ("1830") falls within the period when Żydowo was a town. The town charter was granted in 1752. The significant development of this area dates back to the turn of the 18th and 19th centuries, mainly due to advances in agriculture, trade and crafts. This was facilitated by the economic ties with nearby Gniezno [34]. Textile manufacturing played a special role in this. The aforementioned location played an important role in shaping the town, as Żydowo developed along the main road leading from the north (Gniezno) to the south (Września). According to Weymann [35], it was part of the corridor that connected Silesia with the Baltic Sea. This road, which ran through the town, had the shape of an elongated market square (today, the streets "Plac Obrońców Żydowa") (Figure 4).

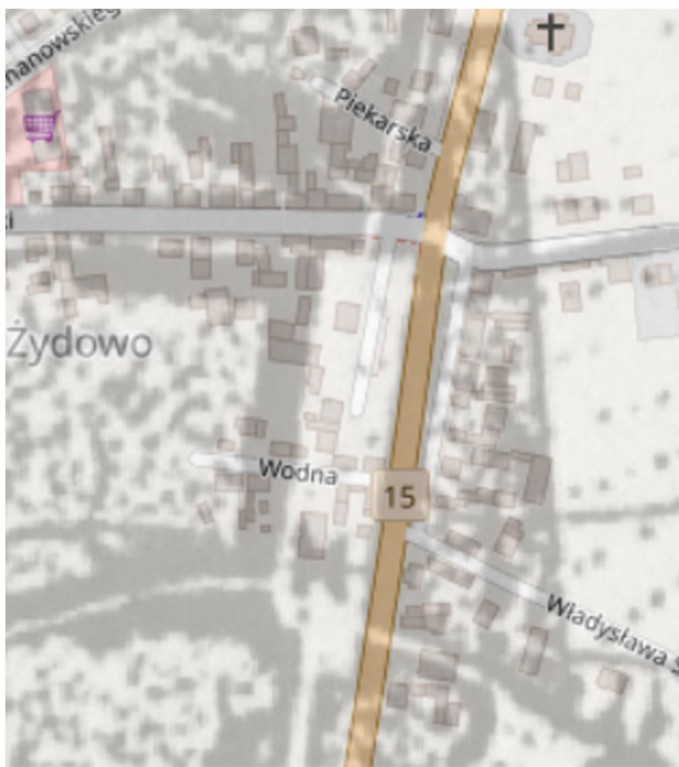

**Figure 4.** The main road transforming into an elongated square as of 1830 and as of 2023.

Apart from the aforementioned market square, around which the buildings were located, Żydowo has similar elements to Niechanowo in terms of its townscape. An important communication axis was the street running to the west, at the end of which there was a residential complex with a park. There were individual houses along the road. To the west, a farm adjoined the estate grounds, and to the south, there were gardens with ponds connected by a tributary of the "Wrześnica" (on a 19th century map, the river was called "Mała Wrześnica"). This stream was fed by waters that flowed through Żydowo from the east and northeast and merged in the center of the town. The 1830 map shows the intersections of roads and watercourses, including in the area of the main square (market square). The area of the town and its immediate surroundings was not forested and was dominated by flat areas with slight differences in terrain.

There were individual buildings along the road that runs parallel to the stream on the southern side. However, neither the road nor the buildings mentioned survived in the landscape over the following decades. Apart from the sections mentioned above, the roads were mostly winding country lanes. An important feature of the topographical landscape of the time were the windmills on the hills on the southern side. The occurrence of this object was particularly pronounced in the eastern part of the town. Descriptive sources confirming the existence of windmills also mention inns and stores in the town. Moreover, an important event in the history of the town was the fire of 1811, which reached the farm and destroyed 12 houses, a manor house, a church and a distillery [34]. Most of the buildings were rebuilt, but this was a factor that influenced the landscape of early 19th century Żydowo.

Changing the selection of the layer to 1888 shows the road network with regulated (straightened) routes. Apart from the fact that such activities were actually carried out on the ground at that time, the specificity of the compared maps should also be taken into account. The older map from the first half of the 19th century was not fully cartometric, and as a result, some of the data recorded on the map did not always correspond to the accuracy of the 1:25,000 scale; for this reason, some topographical features were deformed. The aforementioned road with buildings, which ran along the stream in the central part of

the town and was an extension of the road to Drachów, disappeared from the landscape. In addition, several sections of unpaved roads north of Żydowo also disappeared. In the topographical landscape of the second half of the 19th century, there were signs of two brickworks, one near the farm and the other on the main road leaving the town in the direction of Gniezno. Further along this route, the location of the cemetery is marked, which was already marked at this point on a map from 1830. A very important part of the cultural landscape in the second half of the 19th century was a single-track railroad line opened in 1878, which ran along the western side of Żydowo and which connected Gniezno with Silesia. However, the condition recorded on the 1888 map shows no changes in connection with the development of the areas around the station.

The year 1869 was the moment in the history of Żydowo when it lost its city rights and was given the status of a village. At that time, it was already considered the smallest town in the Prussian division founded under Magdeburg law. According to the literature, the estate was extended in the 1860s, and new buildings were erected on the farm. However, the area of the main square was slightly reduced due to the staking out of a new plot of land [34]. Looking only at the collected cartographic materials from the 19th century, it is difficult to identify negative spatial changes that would indicate a weakening economic position of Żydowo. The landscape changes mentioned above, such as the construction of a railroad line, the expansion of farming or the improvement of the road network, do not seem to correlate with information about the decline in population or the decline in the importance of textile manufacturing in the region.

The landscape recorded on the "1911" layer shows the construction of the railroad station building in Żydowo. There are no longer marked brickyards, and a new facility appeared, "Zgl.", that was surrounded by several buildings not far from the tracks north of the town. In addition, the grade of one of the roads was raised, and a ditch was marked, through which a short stream flowed into the village from the south. Other fragments of watercourses in neighboring areas were highlighted in the same graphic way. In general, the spatial structure or the scope of content had not undergone any changes that might be visible at the scale of the map.

An important landscape feature that can be seen on the 1934 map is the narrow-gage railroad line, which ran off the main railway line to the east and which connected Żydowo, Cielimowo and Gurowo. It ran alongside the farm and circled the village from the south. It was mainly used to transport goods between the farms. According to descriptive information, it was in seasonal use [34]. The rest of Żydowo's landscape remained unchanged compared to 1911.

The record of the state of the environment in a study published 30 years later ("1963") no longer contains any objects associated with the narrow-gage railway line, although according to the literature, the remains of the former line are still visible in the field [34]. Among the objects significantly associated with the 19th century cultural landscape, two windmills that were located on the eastern side of the main road near the market square have disappeared (Figure 5). Their location on a hill is unchanged in the context of the period under consideration from 1830 onwards. A new road was laid out on the north side of the courtyard, along which several buildings were arranged symmetrically. The created functionality of the site, which allows the insertion of later time layers, shows relatively few changes between the period 1911–1963. The transformations mostly affected individual objects. The map shows a new way of developing the market square, in which the main road axis was separated from the rest of the square over a large area.

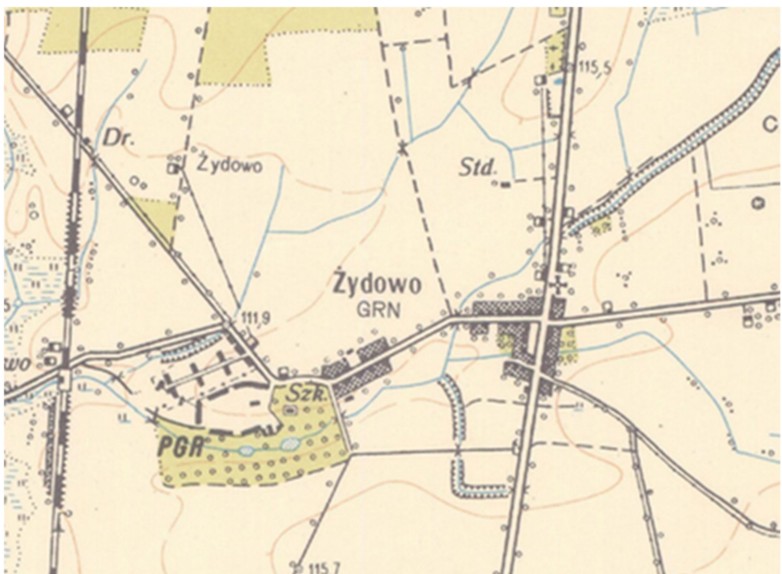

**Figure 5.** Żydowo in the 1960s.

A new period related to the urban changes in the city began at the end of the 20th century ("1977" layer). The composition of the courtyard changed, reconstructions took place and multi-storey blocks appeared on the landscape in place of the earlier buildings (Figure 6). The continuing economic boom was the time when state farm complexes (English for Państwowe Gospodarstwa Rolne = PGR) were built on the sites of the former farms. According to Borowiak [34], this period saw a considerable increase in built-up areas. A stadium, a swimming pool, a cultural center with an auditorium, a health center and a new school were built. A new housing estate was built in the area between the railroad station and the former market square. The changes mentioned above can be seen on the map from 1977. You can also see new buildings on the roads leading out of the village, including along the road leading east toward Gurów and Niechanowo. Due to the progressive spatial development of the town, the rank of individual road sections had changed, e.g., the category of the road running in the southern part of the town (Okrężna Street) had been raised (Figure 6).

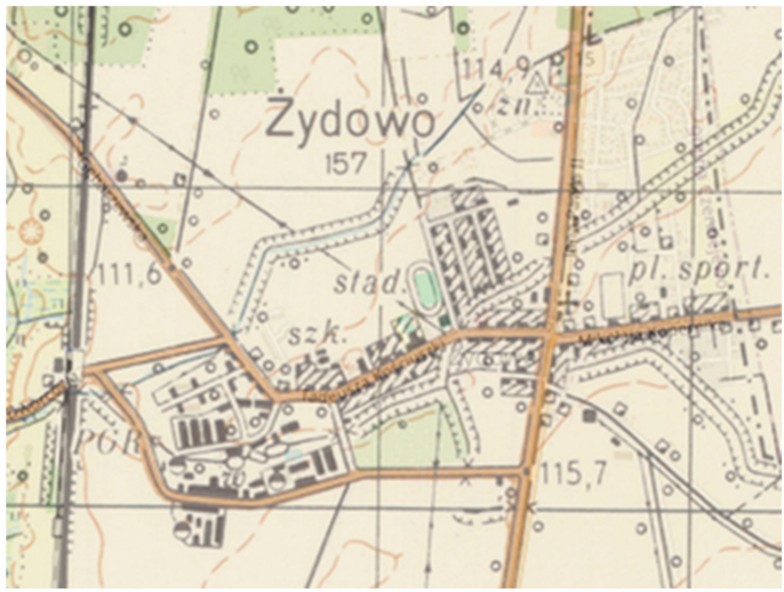

**Figure 6.** Żydowo in the 1970s.

The modern layout of the village ("2023") is characterized by the development of areas on the eastern side of the Gniezno–Września road compared with the situation in 1977. In the northeastern part, a completely new residential area was created, where streets were built between the stream flowing to the center of Żydowo and the green areas to the north. There has also been a visible increase in the built-up area in the areas of the new housing estate, which was built in the previous study period. A dozen or so new residential buildings were also built on the northern side of the road that runs alongside the farm in the direction of the railroad station (Słoneczna Street).

By switching the individual time layers in the created web service, you can obtain a permanent picture of the structure of the town in the 19th century. Even the construction of an important railroad line of supra-regional significance had no influence on the direction of development and spatial changes in Żydowo. The change in the cultural landscape can only be seen in individual topographical objects whose presence in the landscape was not constant for certain decades, e.g., the narrow-gage railway line, windmills, brickworks. It was only in the second half of the 20th century that the use of previously undeveloped land began to change, and this process is progressing gradually. Throughout the period under study, from 1830 onwards, the town remained within the same spatial framework, and it is difficult to detect changes in the landscape related to the change in administrative status in 1869.

### 3.3. Czerniejewo

Of the three administrative units considered in this study, Czerniejewo was an example of an urban area. It was only during the Second World War that the occupying forces decided to combine the area with the territory of the rural municipality. The exact date of the settlement of the town under Magdeburg law is not known, but it is said that it may have taken place before 1390. It is therefore an example of a noble town typical of this period in Greater Poland and Kujawy [36].

The landscape marked on the 1830 map shows a more extensive structure related to roads than in the case of Żydowo and Niechanowo (Figure 7). The town was also characterized by its larger area, and the fact that it had city rights is expressed on the map by the graphic features of the letters. According to the legend of 19th century Prussian maps, the names of cities and towns were written in capital letters. In this context, it can be seen in Figure 7 that Żydowo was incorrectly drawn on the map from the first half of the 19th century because it also had the status of a town at that time. The similarity between the three analyzed places is reflected in the town layout, which is characterized by an elongated shape based on the main axis of communication and ending with a manor house area. In the case of Czerniejewo, the palace from 1780 was located in the northern part of the town. A wide road lined with buildings led to it, and at the other end of the road was a church. The buildings were located on almost all roads leading out of the town, and their particular concentration is visible in the southern part, where the street had the shape of an irregular square.

The locations of windmills were marked on the map, indicating the agricultural character of this area, which remained unchanged in later periods. Four objects were marked on the southern outskirts of the town, one on a hill near the fork in the road on the western side of the town and one in the north-eastern part, on the tributary of the Wrześnica. In addition to six windmills, three water mills are also mentioned in the literature, but they are not shown on the map [36]. The landscape of the town is also characterized by a relatively varied network of surface waters. The main meridional river, Wrześnica, passed to the right of the town and was accompanied by wetlands that marked the natural border of Czerniejewo. Two smaller tributaries flowed through the town from the west, and their waters rose to form ponds. A larger one was located in the area of the palace and several smaller ones in the central part of Czerniejewo. There were no forests in the immediate vicinity of the town, but on the western side, there was an extensive complex ("JEZIERCER CZERNIEJEWER FORST") at a distance of about 2 km.

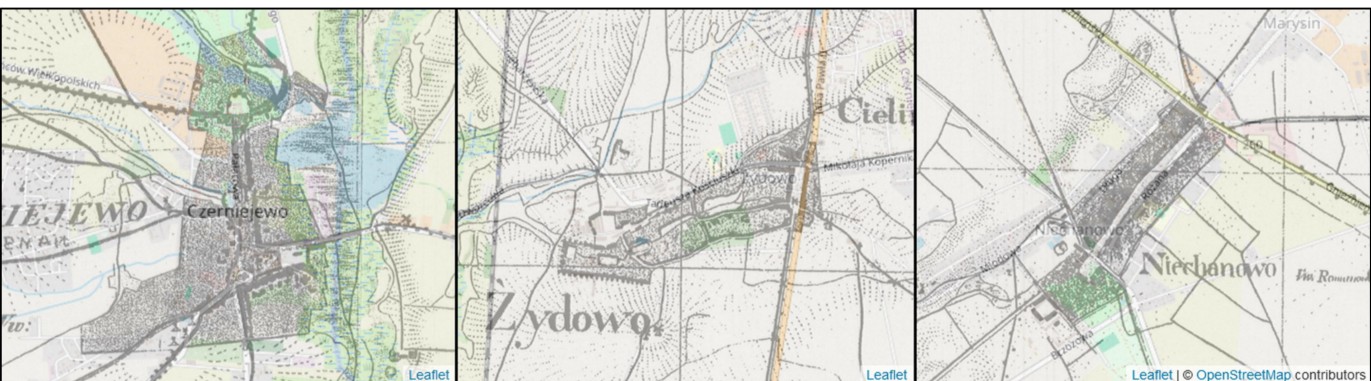

**Figure 7.** Cultural landscape of three areas in 1830 against the background of contemporary data.

The state of the area shown on a map published over 50 years later ("1888") shows a number of changes that had occurred in the city's landscape. Most of these concerned individual topographical features that did not disrupt the earlier settlement structure. One of the few elements that led to the present spatial arrangement is a section of a wide, tree-lined road that was laid out in the northwestern part between the manor house area and one of the tributaries. New buildings were also constructed along the road. Another new section of the road ran along the western boundary of the manor park, next to the farm buildings. Inside the surrounding park, however, the course of paths and a significantly enlarged area of ponds can be seen (Figure 8). A few hundred meters away from the palace, to the northeast, there was a pheasantry ("Fasanerei"). A brick building is marked at this location on a map from 1830.

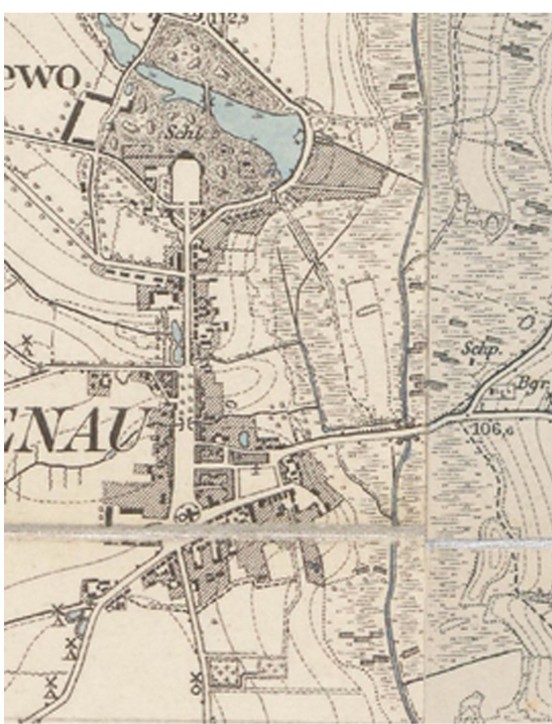

**Figure 8.** Changes in the landscape of Czerniejewo in 1888.

The water flowing away on the east side of the town was diverted into a new bed along the road, where it flowed into the main river before the bridge. In this way, a bridge closer to the town disappeared from the landscape. At the fork in the road on the other side of the river, the location of the cemetery is indicated, and a windmill has appeared. The second cemetery was located on the main road toward Września. The map shows the development

in the southeastern part. You can also see the straightening of the previously existing roads and the regulation of the shape of the square, where rows of trees have appeared. The cultural landscape of the second half of the 19th century is also complemented by a brickyard at the northern end and gravel pits on the western side.

The railroad road built in 1878 was also a very important element. It was the same route that led through Żydowo; Czerniejewo was the next station. However, in the case of this town, the station was located about 6 km from the center and could be reached via the only road leading out of the town to the east. The creation of a railroad connection, which was an important feature of the economic changes in the second half of the 19th century, had no impact on the transformation of the urban landscape and its surroundings in this case [13,37].

The most important difference in the landscape recorded on the next map published after more than 20 years was the appearance of the narrow-gage railway line. Its tracks branched off the main railroad line from the station in Czerniejewo and connected individual farms, reaching the town from the southeast. The map shows the town with a route that led through the river and the wetland on which a dam was built. After crossing the main axis of the town, the train continued along the western boundary of the manorial area for another half kilometer to the north along the road. Changes related to the emergence of a new means of transportation are visible in the southeastern part, even before the river. A new brickworks was built near the marked path, which was connected to the road by a railway siding. This area was also connected to the main road, where new buildings were constructed on the farther section. The second brickworks in the town's landscape was built on the southern edge of the town. The location of the brickworks near the palace was slightly changed, and the new facility was marked by a pheasantry, together with several larger buildings. Among the other facilities, it is noticeable that the number of windmills had decreased compared to previous years.

The layer created on the basis of a map from 1934 shows the already shortened route of the narrow-gage railroad, which ends at the courtyard of the palace. All the markings of the brickworks as well as the associated buildings and the siding in the southeastern part have disappeared from the topographical landscape. The areas bordering the courtyard from the west are characterized by new buildings. There were also changes in the park, where an additional canal was dug for the watercourse that fed the palace ponds, and two parallel roads were laid out. The residence was also rebuilt in the first decades of the 20th century, including the extension of the eastern part and an outbuilding [36]. The area of woodland surrounding this part of the town also increased.

A significant change in the landscape took place on the eastern side of the city. Part of the wetland was transformed into an extensive pond, which is labeled as a lake on the map. On the southern side, the reservoir bordered a large sawmill yard. On the southern edges, however, further buildings were erected to fill the spaces at the forks in the road. There was also a steam mill among the buildings.

Another time layer from 1963 shows the use of previously undeveloped areas in the southwestern part of the city. The structure of the castle park and its surroundings was redesigned. This space was fragmented by the marking of roads and smaller paths, and a new section of the main road was built from the northeast toward the north. The Wrześnica tributary, which was routed along the road in a new riverbed on the 1888 map, flowed closer to the city center again under the second bridge in the second half of the 20th century. The narrow-gage railway line running nearby was shortened and ended at the sawmill. New farms were also built along the main road leaving the town to the east.

During the Second World War, Czerniejewo was part of a rural municipality by order of the occupying forces, but after the end of the fighting, its proper status was restored. In the middle of the 20th century, consideration was given to revoking its city rights due to its declining importance. The main factors always included the lack of developed industry and the distance from the main railroad line. There were only small businesses processing agricultural products. In addition, the town still did not have water and sewage

systems, had difficulties with the electricity supply and the overall financial situation was not favorable. Despite the above-mentioned unfavorable factors, the town retained its status [20,36].

Over the next dozen or so years, until 1977, the narrow-gage railway line, which was a connection to the main line outside the town, was dismantled. All that remained was a short section of unused track on the southeastern outskirts of the town. A new water reservoir was created in the landscape, located in the existing wetlands on the eastern bank of the Wrześnica River. Visible changes in the urban structure also included new buildings on the western side of the town and along the road leaving the town to the southwest. The former farm next to the manor house, similar to the one in Żydowo, retained its function and became a unit of state farms. In the 1970s, there was a general improvement in economic development. Mechanization and the use of fertilizers in crops became popular in agriculture. The number of inhabitants who found work in agriculture, trade and the service sector slowly increased [36].

The inclusion of the layer with the current state ("2023") shows that the direction of urban development begun in the 1970s continued in the following decades. The current spatial structure shows the further development of settlement processes toward the southwest. A new element in the landscape is a large housing estate with a network of new roads built on the western side. A much smaller residential area was also created in the northeastern part of the town, bordering the river. Buildings were erected here in earlier decades, and it is now a housing estate. In general, the settlement network of Czerniejewo has become denser, often on the basis of newly marked streets (e.g., Osiedle Papieża Jana Pawła II).

According to the literature, the town was an important center in the region throughout its history, which, like many others, developed in the 18th century due to the flourishing of textile manufacturing. An important feature that influenced its shape was the lack of walls or city gates—it was open. Another important event in its history was a great fire in the 18th century [36]. The main obstacles to development were the great distance from the railroad station and the lack of developed industry. Despite this, the town gradually changed over the years and maintained its status (with the exception of the war period). Compared with the other two areas, the landscape of Czerniejewo has changed the most, but the core of the town from the first half of the 19th century has remained unchanged. This is reflected in the literature but even more so in the content of individual maps.

## 4. Conclusions

In the case of Czerniejewo, it can be seen that the town charter, the location of the manor house and the adjacent farm were factors that attracted new people to the town. A comparison of the three areas shows that the city rights and the associated administrative tasks were not without significance for the spatial development and transformation of the cultural landscape. This is also confirmed by the history of the nearby Żydowo town. Even though Żydowo, like Czerniejewo, had an estate and a farm and was located on the main transportation routes (road and railroad), it lost its town charter (city rights). Then, for many decades, the spatial structure of the town did not develop significantly beyond the 19th century layout.

Throughout the entire period under study, Niechanowo remained a village, even though its structure also included a manorial area with a farm and a narrow-gage railway connection with other centers (including Gniezno). Compared with other areas, this village is an example of the preservation of many features of the pre-industrial landscape (cultural landscape of the first half of the 19th century). This is mainly due to the lack of industrial development and the fact that it did not fulfill any administrative functions for the region, which was an important factor in the development of many towns.

The city's status (city rights) was one of the factors that contributed to the development and transformation of the settlement units. Czerniejewo did not lose its city rights, and its development proceeded gradually over the years, while Żydowo, which was deprived of

its town status, stood still in its spatial development. It should be noted that in the case of Żydowo, the change in town status was not only the result of existing processes connected with the new situation after the Congress of Vienna, but it was also connected with the economic changes in the 19th century. According to the literature, the loss of town status was due to the declining importance of textile manufacturing in the region and a strong population exodus. Perhaps retaining its town status would have accelerated and increased the extent of the spatial changes in the ensuing years.

Despite the differences in the status of the towns, their common features include the following characteristic elements of the cultural landscape of 19th century Greater Poland: an urban road leading to the manorial areas (castle with a park), the functioning of farms near the manorial areas, the presence of windmills and sand and gravel quarries and the construction of narrow-gage railroads, which, at the turn of the 19th and 20th centuries, formed a highly developed network characteristic of the region under study.

By combining archived source material and modern map programming tools, it was possible to create a platform for researching the former cultural landscape and its ongoing transformation. Thanks to the options for changing the individual time layers, the possibility of moving and scaling maps and options for changing the transparency, it was possible to trace the development of individual areas using selected objects or spatial structures.

The authors are aware that, in practice, the limited human ability to analyze and integrate information from maps for larger areas can be a problem. Researchers are not able to extract large amounts of data in a reasonable amount of time [38]. For this reason, heritage experts and data scientists are exploring the use of technology to bridge the gap between manual labor and the automatic recognition of spatial features in historical maps. There are two main groups of methods for recognizing and extracting information from historical maps. The first method uses neural networks [6,39–41]. Unfortunately, this method is clearly not effective for old maps with complex styles. Moreover, this method relies on the assessment of the global similarity of stylized images and integration with satellite imagery to produce a neural output image rather than performing feature detection tasks [42]. The second group of methods includes geographic information system (GIS) integration [43]; in particular, the object-based image analysis (OBIA) method [44,45]. Most work on geographic feature recognition has found that success is largely influenced by the characteristics of old maps. In particular, the way spatial elements are represented in older cartographic images is very different from modern maps. The authors of this article believe that a traditional cultural landscape analysis based on archival maps will still be an effective research method, but for larger areas, it is supported by the latest technologies.

Furthermore, the authors see the possibility of enriching the research on the transformation of the former landscape with the method of landscape metrics. This involves examining changes using indicators that illustrate various landscape features, e.g., composition and spatial configurations. One example is the study of forest landscapes in Turkey [46]. Records were used to observe the changes mainly caused by human activities and the course of secondary succession over a period of about 30 years. This method was also used to study changes in woodlands area in the Puszcza Zielonka Landscape Park (western Poland) in the period 1830–2010 [47]. These studies, based on the use of cartographic archival material, provided quantitative and qualitative characteristics of changes in the area and in the spatial arrangement of forest areas.

**Author Contributions:** Conceptualization, Dariusz Lorek; methodology, Dariusz Lorek; software, Tymoteusz Horbiński; formal analysis, Dariusz Lorek; resources, Dariusz Lorek and Tymoteusz Horbiński; data curation, Tymoteusz Horbiński; writing—original draft preparation, Dariusz Lorek and Tymoteusz Horbiński; writing—review and editing, Dariusz Lorek and Tymoteusz Horbiński. All authors have read and agreed to the published version of the manuscript.

**Funding:** This research received no external funding.

**Data Availability Statement:** Not applicable.

**Conflicts of Interest:** The authors declare no conflicts of interest.

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
