# Peer review of "Changes in the 19th Century Cultural Landscape with Regard to City Rights in Western Poland"

_ijgi, doi:10.3390/ijgi13050164_

Round 1

Reviewer 1 Report

Comments and Suggestions for Authors

Methodology and outcomes are coherent with the research's purpose. Background and topics are well presented and the interactive map is accessible for the overall evaluation.

Author Response

Reviewer 1  

Methodology and outcomes are coherent with the research's purpose. Background and topics are well presented and the interactive map is accessible for the overall evaluation. 

Thank you for your comment. 

Reviewer 2 Report

Comments and Suggestions for Authors

This paper describes some interesting background work: it analyses the transformations of urban space that took place in three different (and very close one from each other) locations and relates them to changes in administrative category (or the absence of such changes in one case). But the paper offers little more than this shallow analysis. No further contribution can be found in the conclusions, either. I'm sure the work can be interesting to record, probably in a Digital Humanities publication, but the IJGI forum doesn't seem to be the most appropriate.

I believe that the content of this article doesn’t fit in the scope of the IJGI publication. Surely there are other publications for this article to fit in an appropriate way. Looking through the aim and the various sections of the journal's csope, we cannot see  posible contribution of this paper, so I regret to say that I do not recommend its publication.

However, I would like to point out some issues mainly addressed to the authors.

First, about the title of the paper. The title is misleading. In fact, I confess that I wanted to check the article for the presence of the words “thematic web map” in its title. But there is no trace of any thematic map in the text. It shows that the term “thematic map” is not known by the authors, since the maps used for the analysis are topographical. I would rewrite it as "using old maps" or "using old topographic maps".

The web application presented seems very poor to carry out the proposed analysis and brings nothing new to other map comparators. Some issues with it:

- You have to look at all three cities at the same time. I’d like to have the possibility of an individual analysis too.

- I don’t find it adecuate limiting the comparison of the maps against to one only source; the OSM map. The possibility of superimposing two different old maps is not given, and it should be offered as it is of main interest when carrying out the proposed analysis.

- I don’t see the seleccion of ths map, the OSM. It doesn't seem to be the most suitable option for comparing landscapes, either natural or urban. I'm not from Poland but due to the quality of Polish cartography I guess it's possible to overlay with a richer topographic map than OSM.

The scalability of the application is not foreseen. Then, the very title of the application, of this “Thematic Web Map” (I repeat; inadequate denomination) does not seem to be in line with what it offers. It should include the place names of the cities subjected to the analysis (e.g. as a subtitle)

I suggest that the authors consider the possibility of using a StoryMap format instead of this viewer for the dissemination of their analysis and conclusions. I think it would be more useful for those interested in following their explanations.

Best regards

Author Response

Reviewer 2 

This paper describes some interesting background work: it analyses the transformations of urban space that took place in three different (and very close one from each other) locations and relates them to changes in administrative category (or the absence of such changes in one case). But the paper offers little more than this shallow analysis. No further contribution can be found in the conclusions, either. I'm sure the work can be interesting to record, probably in a Digital Humanities publication, but the IJGI forum doesn't seem to be the most appropriate. 

The authors disagree with the view that “the paper offers little more than this shallow analysis”. The authors used a cartographic research method (Saliszczew, 1998) in which maps were the basic source of information. Therefore, the degree of accuracy (detail of information) is related to the scale of the cartographic material used. The chosen procedure was also based on methodological principles for the use of old maps in research (Buczek 1966, Scharfe 1990). Accordingly, the information obtained was verified and supplemented using information from other (descriptive) source studies. 

Saliszczew K.A. Kartografia ogólna, Wydawnictwo Naukowe PWN, Warszawa, Poland, 1998. 

Buczek, K. History of Polish Cartography from the 15th to the 18th Century; ZakÅ‚ad Narodowy im OssoliÅ„skich: WrocÅ‚aw, Poland, 1966. 

Scharfe W. Kartographiegeschichte – Grundlagen – Aufgaben – Methoden, 4. Kartographiehistorisches Colloquium Karlsruhe 1988: Vorträge und Berichte, Dietrich Reimer Verlag, Berlin, Germany, 1990, pp. 1–10. 

“No further contribution can be found in the conclusions, either.” 

In response to the question about the further contribution, the authors point out that they have examined the situation from the past and related it to the current situation - with this approach, the research process should be considered complete (finished). 

The authors see a field for further research in developing this type of research from a regional perspective and not just in relation to a single city. Among other things, this makes it possible to consider topics of broader significance, such as the aspect of city rights, which is discussed in this research. In addition, the authors see the possibility of further developing the map service, particularly with additional interactive elements and taking into account the quantitative and qualitative information from the analyzes carried out. 

Furthermore, the research is a continuation of earlier work (SmaczyÅ„ski et. al 2022, Lorek and HorbiÅ„ski 2020, HorbiÅ„ski and Lorek 2020). Against this background, the research process described in this article represents both a confirmation and a further development of the methodological assumptions made and developed to date. 

SmaczyÅ„ski, M.; Lorek, D.; Zagata, K.; HorbiÅ„ski, T. Cultural Heritage with the Use of Low-Level Aerial Survey Techniques, Space Modelling and Multimedia Reconstruction of the Topographic Landscape (Example of a Windmill in Western Poland). KN J. Cartogr. Geogr. Inf. 2022, 72, 279–291. https://doi.org/10.1007/s42489-022-00122-6 

Lorek, D.; HorbiÅ„ski, T. Interactive Web-Map of the European Freeway Junction A1/A4 Development with the Use of Archival Cartographic Sources. ISPRS Int. J. Geo-Inf. 2020, 9 (7), 438. http://dx.doi.org/10.3390/ijgi9070412 

HorbiÅ„ski T.; Lorek D. The use of Leaflet and GeoJSON for creating the interactive web map of the preindustrial state of the natural environment. Journal of Spatial Science, 2022, 67(1), 61-77. http://doi.org/10.1080/14498596.2020.1713237 

Furthermore, the authors see the possibility of enriching research on the transformation of the former landscape with the method of landscape metrics. This involves examining changes using indicators that illustrate various landscape features, e.g. composition and spatial configuration. One example is the study of forest landscapes in Turkey (Terzioglu et al. 2009). Records were used to observe the changes mainly caused by human activities and the course of secondary succession over a period of about 30 years. This method was also used to study changes in the area of woodlands in the Puszcza Zielonka Landscape Park (western Poland) in the period 1830–2010 (Macias et al. 2022). These studies, based on the use of cartographic archival material, provided quantitative and qualitative characteristics of changes in the area and spatial arrangement of forest areas.  

TerzioÄŸlu S.; BaÅŸkent E.Z.; KadıoÄŸulları A.I. Monitoring forest structure at landscape level: a case study of Scots pine forest in NE Turkey. Environ. Monit. Assess. 2009, 152, 71-81. https://doi.org/10.1007/s10661-008-0297-3   

Macias, A.; Bródka, S.; Kubacka, M. Zmiany powierzchni leÅ›nych na terenie Parku Krajobrazowego Puszcza Zielonka w ostatnich 180 latach w aspekcie krajobrazowym. Badania Fizjograficzne Seria A - Geografia Fizyczna, 2022, 13(A 73), 123–134. https://doi.org/10.14746/bfg.2022.13.7 

I believe that the content of this article doesn’t fit in the scope of the IJGI publication. Surely there are other publications for this article to fit in an appropriate way. Looking through the aim and the various sections of the journal's csope, we cannot see  posible contribution of this paper, so I regret to say that I do not recommend its publication. 

The article represents a methodological evolution of our previous articles published, among others, in this journal. We believe that we recognize the potential contribution of this work to this journal. 

The interdisciplinary nature of the topics covered and the methodology used are in line with the objectives of the journal (Spatial analysis and Cartography), although we are aware that, due to the above-mentioned interdisciplinary nature, it could also be submitted to other journals (Journal of Special Science, KN, etc.). 

However, I would like to point out some issues mainly addressed to the authors. 

First, about the title of the paper. The title is misleading. In fact, I confess that I wanted to check the article for the presence of the words “thematic web map” in its title. But there is no trace of any thematic map in the text. It shows that the term “thematic map” is not known by the authors, since the maps used for the analysis are topographical. I would rewrite it as "using old maps" or "using old topographic maps". 

We agree with the above comment and have therefore decided to change this. The source used was, as correctly noted, topographic maps. 

The thematic map is understood here less as a source of information than as a product of the activities carried out. By developing the website on the basis of the collected maps, a specific thematic area was covered, namely the importance of city rights in the context of the development of the places and the impact on the transformation of the landscape. The research was carried out in the context of selected (thematic) objects and spatial structures. 

The web application presented seems very poor to carry out the proposed analysis and brings nothing new to other map comparators. Some issues with it: 

- You have to look at all three cities at the same time. I’d like to have the possibility of an individual analysis too. 

- I don’t find it adecuate limiting the comparison of the maps against to one only source; the OSM map. The possibility of superimposing two different old maps is not given, and it should be offered as it is of main interest when carrying out the proposed analysis. 

According to the idea of the article, the aspect of simultaneous analysis of maps for three places was crucial in the context of the study and visualization of the conclusions. The changes that took place in certain areas over the course of 200 years, although very important, represented secondary information (background). The authors therefore did not foresee the possibility of such a comparison. 

- I don’t see the seleccion of ths map, the OSM. It doesn't seem to be the most suitable option for comparing landscapes, either natural or urban. I'm not from Poland but due to the quality of Polish cartography I guess it's possible to overlay with a richer topographic map than OSM. 

The methodological assumptions we developed earlier included OSM as a reference map to capture spatial changes. 

We agree with the opinion that there are other cartographic studies, including landscape maps, but at other (smaller) scales. On the other hand, the current BDOT10k database also exists, but its level of detail corresponds to a scale of 1:10,000. Its use would therefore require a generalization, which is always subject to a certain degree of subjectivity. Therefore, after careful examination of the existing studies, a general geographic database was selected that is close to global standards in terms of accuracy level and whose content scope corresponds to the assumed accuracy level and correlates with the content of the selected archive maps. 

- The scalability of the application is not foreseen. Then, the very title of the application, of this “Thematic Web Map” (I repeat; inadequate denomination) does not seem to be in line with what it offers. It should include the place names of the cities subjected to the analysis (e.g. as a subtitle) 

In the authors' opinion, the inclusion of the city names in the title will not affect the identification of these areas for readers from around the world, although we have included the names in the title of the article at the suggestion of the reviewer. 

I suggest that the authors consider the possibility of using a StoryMap format instead of this viewer for the dissemination of their analysis and conclusions. I think it would be more useful for those interested in following their explanations. 

In our research, we consider open source solutions (for both data and technology) that do not burden the user with specific set of themes.  

Reviewer 3 Report

Comments and Suggestions for Authors

The way the history and cartography are connected in this article is particularly interesting. The comparison of the growth of three Polish towns serves as the main theme.

I believe there should have been more information about the underlying data processing (maps) in section 2.2. Specifically, how they are georeferenced to allow comparison between the individual maps. References to the literature on georeferencing historical maps would be beneficial, as would a brief explanation of the georeferencing process.

It would be helpful to discuss the benefits of a web-based historical landscape information system in more detail in section 2.3, along with some relevant literature.

Regarding the data analysis, it would be beneficial to note that in contrast to the application's visual comparison, the vectorized map layers (planimetry and elevation) can also be compared, yielding qualitative results.

Despite the aforementioned criticisms, the article is interesting particularly when viewed from the historical perspective of Polish town development. Neither the paper's originality nor the way it was presented (as an online map application) is particularly impressive. However, I believe the results that have been presented are interesting, particularly for historians who are interested in the history of town development.

Author Response

Reviewer 3 

The way the history and cartography are connected in this article is particularly interesting. The comparison of the growth of three Polish towns serves as the main theme. 

I believe there should have been more information about the underlying data processing (maps) in section 2.2. Specifically, how they are georeferenced to allow comparison between the individual maps. References to the literature on georeferencing historical maps would be beneficial, as would a brief explanation of the georeferencing process. 

We have included the reviewer's recommendations on georeferencing in section 2.2 

It would be helpful to discuss the benefits of a web-based historical landscape information system in more detail in section 2.3, along with some relevant literature. 

Following the reviewer's recommendations, we have expanded section 2.3 to explain the advantages of the map service in more detail. 

Regarding the data analysis, it would be beneficial to note that in contrast to the application's visual comparison, the vectorized map layers (planimetry and elevation) can also be compared, yielding qualitative results. 

Thank you for your comment. We agree with the possibility of activity within vector layers (we have previously published studies of this type; SmaczyÅ„ski et.al 2022, Lorek and HorbiÅ„ski 2020, HorbiÅ„ski and Lorek 2020). However, the methodology presented in this article is based on the use of old maps as a carrier of comprehensive (complete) information about the landscape. 

In this context, the article can be seen as a continuation of previous work. The described research process is both a confirmation and a further development of the previously assumed and developed methodological assumptions.  

Despite the aforementioned criticisms, the article is interesting particularly when viewed from the historical perspective of Polish town development. Neither the paper's originality nor the way it was presented (as an online map application) is particularly impressive. However, I believe the results that have been presented are interesting, particularly for historians who are interested in the history of town development. 

According to the authors, originality is shown in the consideration of the landscape of settlement units, taking into account an important criterion, namely "town rights". This way of approaching the subject broadens the scope of research from individual towns to the level of the entire region. In this context, the map service is an important tool for comparing, analyzing and presenting the results. It is useful to refer to the earlier publications cited above when developing the current methodology. 

Round 2

Reviewer 2 Report

Comments and Suggestions for Authors

Dear authors,

After this second review, it is my opinion that, while the content is valuable, this article doesn't fit the focus of IJGI; I believe that the content expected in this journal is of a different nature. Furthermore, the section introduced on the map service lacks depth, novelty and relevance.

I therefore reiterate my position that I would advise against accepting this article for publication in your journal and would suggest the authors to explore alternative avenues for its dissemination.

Author Response

Thank you for your comment.